# Long-Term Survival Outcomes of Cytoreductive Nephrectomy Combined with Targeted Therapy for Metastatic Renal Cell Carcinoma: A Systematic Review and Individual Patient Data Meta-Analysis

**DOI:** 10.3390/cancers13040695

**Published:** 2021-02-09

**Authors:** Stepan M. Esagian, Ioannis A. Ziogas, Dimitrios Kosmidis, Mohammad D. Hossain, Nizar M. Tannir, Pavlos Msaouel

**Affiliations:** 1Oncology Working Group, Society of Junior Doctors, 151 23 Athens, Greece; stepesag@sni.gr (S.M.E.); kosmidimi@yahoo.gr (D.K.); 2Surgery Working Group, Society of Junior Doctors, 151 23 Athens, Greece; iaziogas@sni.gr; 3Department of Medicine, School of Health Sciences, National and Kapodistrian University of Athens, 157 72 Athens, Greece; 4Faculty of Medicine, Jalalabad Ragib-Rabeya Medical College, Sylhet 3100, Bangladesh; mdporag@gmail.com; 5Department of Genitourinary Medical Oncology, The University of Texas MD Anderson Cancer Center, Houston, TX 77030, USA; ntannir@mdanderson.org; 6Department of Translational Molecular Pathology, The University of Texas MD Anderson Cancer Center, Houston, TX 77030, USA

**Keywords:** cytoreductive nephrectomy, renal cell carcinoma, targeted therapy, tyrosine kinase inhibitors, VEGF inhibitors, mammalian target of rapamycin inhibitors

## Abstract

**Simple Summary:**

Cytoreductive nephrectomy (CN) refers to the removal of the primary renal tumor in the setting of metastatic renal cell carcinoma. In the past, the combination of CN with cytokine-based immunotherapy was considered the standard of care. However, CN’s role during the targeted treatment era remains controversial. We attempted to address this issue by performing a systematic review and meta-analysis of the literature. We synthesized data from 15 studies comparing CN and targeted therapy to targeted therapy alone. Our results show that CN combined with targeted therapy was associated with increased survival compared to targeted therapy only. Careful patient selection is required to take full advantage of any survival benefit that CN may offer. Future research endeavors should focus on developing appropriate prognostic models to guide appropriate patient selection for CN.

**Abstract:**

The role of cytoreductive nephrectomy (CN) in the treatment of metastatic renal cell carcinoma (mRCC) remains controversial during the targeted therapy era. To reconcile the current literature, we analyzed the reported survival data at the individual patient level and compared the long-term survival outcomes of CN combined with targeted therapy vs. targeted therapy alone in patients with mRCC. We performed a systematic review of the literature using the MEDLINE, Scopus, and Cochrane Library databases (end-of-search date: 21 July 2020). We recuperated individual patient data from the Kaplan–Meier curves for overall (OS), progression-free (PFS), and cancer-specific survival (CSS) from each study. We subsequently performed one-stage frequentist and Bayesian random-effects meta-analyses using both Cox proportional hazards and restricted mean survival time (RMST) models. Two-stage random-effects meta-analyses were also performed as sensitivity analyses. A subgroup analysis was also performed to determine the effect of CN timing. Fifteen studies fulfilling our inclusion criteria were identified, including fourteen retrospective cohort studies and one randomized controlled trial. In the one-stage frequentist meta-analysis, the CN group had superior OS (hazard ratio [HR]: 0.58, 95% confidence interval [CI]: 0.54–0.62, *p* < 0.0001) and CSS (HR: 0.63, 95% CI: 0.53–0.75, *p* < 0.0001). No meaningful clinical difference was observed in PFS (HR: 0.90, 95% CI: 0.80–1.02, *p* = 0.09). One-stage Bayesian meta-analysis also revealed superior OS (HR: 0.59, 95% credibility interval [CrI]: 0.55–0.63) and CSS (HR: 0.63, 95% CrI: 0.53–0.75) in the CN group, while no meaningful clinical difference was detected in PFS (HR: 0.91, 95% CrI: 0.80–1.02). Similar results were obtained with the RMST models. The OS benefit was also noted in the two-stage meta-analyses models, and in the subgroup of patients who received upfront CN. The combination of CN and targeted therapy for mRCC may lead to superior long-term survival outcomes compared to targeted therapy alone. Careful patient selection based on prognostic factors is required to optimize outcomes.

## 1. Introduction

Over the past year, almost 74,000 new cases of kidney and renal pelvis cancer were recorded in the United States alone [1]. Despite the advances in imaging modalities allowing for earlier diagnosis, a significant proportion of patients (>10%) present with metastatic disease [2]. The management of metastatic renal cell carcinoma (mRCC) has undergone a significant shift over the past three decades. The discovery of RCC’s immunogenicity led to the establishment of interleukin-2 and interferon-alpha as first-line treatments for mRCC during the 1990s, marking the so called “cytokine era” [3,4,5]. Cytoreductive nephrectomy (CN) refers to the removal of the primary renal tumor in the metastatic setting and met with sporadic success during the 20th century [6,7]. In the early 2000s, CN re-emerged and its combination with cytokine-based immunotherapy became the new standard of care, following the results of two randomized controlled trials (RCTs) [8,9]. Subsequently, several RCTs established the superiority of targeted therapies over cytokine-based immunotherapy [5,10,11,12], leading to a paradigm shift and the new “targeted therapy era” in the treatment of mRCC. The implementation of CN declined along with cytokine-based immunotherapy, mainly because its role when combined with targeted therapies remained unclear [13].

Over the last decade, multiple large retrospective studies have shown promising results with the combination of CN and targeted therapies [14,15]. In contrast, the Cancer du Rein Metastatique Nephrectomie et Antiangiogéniques (CARMENA) trial reported non-inferiority of the targeted therapy sunitinib alone vs. CN followed by sunitib in the intention-to-treat (ITT) population [16]. However, non-inferiority trials, such as CARMENA, have in practice a greater than 80% probability of reaching a verdict of non-inferiority, particularly when protocol adherence rates are low. CARMENA suffered from such low protocol adherence rates, which prevented a full per-protocol analysis of its data, and even a partial per-protocol analysis (termed “PP2” in the CARMENA report) led to inconclusive results [16]. Nevertheless, the results of CARMENA casted doubt upon the value of CN during the targeted therapy era, and a subsequent retrospective report also disputed the long-term benefits of CN [17]. As a result, CN in the setting of targeted therapy remains controversial. Therefore, we sought to systematically review and synthesize the totality of currently available evidence comparing the long-term survival outcomes of CN combined with targeted therapy over targeted therapy alone in patients with mRCC.

## 2. Materials and Methods

### 2.1. Study Design and Inclusion/Exclusion Criteria

This systematic review and meta-analysis was performed according to the Preferred Reporting Items for Systematic Reviews and Meta-Analyses (PRISMA) guidelines and in line with a protocol developed and agreed upon by all authors (Appendix A) [18]. An Institutional Review Board approval or patient written consent was not necessary, as we used already published data.

We applied the Population/Participants, Intervention, Comparison, Outcomes, Study design (PICOS) framework to define the study selection criteria as follows:Participants: Patients of any age, sex, or race with mRCC.Intervention: Cytoreductive nephrectomy (CN), either before (upfront) or after the initiation of targeted therapy (deferred). We defined targeted therapy as systemic therapy with vascular endothelial growth factor (VEGF) receptor-directed tyrosine kinase inhibitors (TKIs) (e.g., sunitinib, axitinib, pazopanib, sorafenib, famitinib), anti-VEGF monoclonal antibodies (e.g., bevacizumab), or mammalian target of rapamycin (mTOR) inhibitors (e.g., everolimus, temsirolimus).Comparison: Targeted therapy without CN.Outcomes: Long-term survival outcomes, including overall survival (OS), progression-free survival (PFS), and cancer-specific survival (CSS) of the intention-to-treat populations, when applicable.

Original randomized clinical trials and non-randomized cohort studies (both prospective and retrospective) comparing the combination of CN and targeted therapy vs. targeted therapy alone for mRCC, published in English, were considered eligible for inclusion. The exclusion criteria were defined as follows: (i) articles without a full text in English; (ii) irrelevant articles; (iii) animal studies; (iv) case reports; (v) narrative or systematic reviews and meta-analyses; (vi) letters to the editor, editorials, commentary, errata, perspectives without any primary patient data; (vii) published abstracts without any available full text; (viii) non-comparative studies (<2 study arms); (ix) studies without any extractable data for the outcomes of interest.

We assessed all eligible studies for overlapping populations based on the author list, study center, country of origin, and dates of patient enrollment. Between studies with overlapping populations, we included those having the largest patient sample, reporting granular data for the outcomes of interest, and providing Kaplan–Meier curves that would permit reconstruction of individual patient data (IPD). However, when data on additional outcomes were provided through multiple studies, we extracted data from all of them. In these cases, we did not sum the populations of each study in the overall subject numbers, as they represented additional analyses on the same cohorts.

### 2.2. Literature Search Strategy

A systematic search was performed using the MEDLINE (via PubMed), Cochrane Library, and Scopus bibliographic databases (end-of-search date: 21 July 2020) by two independent researchers (S.M.E. and M.D.H) using the term “cytoreductive nephrectomy”. All disagreements on article inclusion were resolved after reaching a consensus. In accordance with the snowball methodology, references from all included articles and previously published systematic reviews/meta-analyses were also manually searched to identify any potentially missed but otherwise eligible for inclusion studies [19].

### 2.3. Data Tabulation and Extraction

Data tabulation and extraction for evidence synthesis were performed using standardized, pre-piloted spreadsheets. Two reviewers (S.M.E. and M.D.H.) independently extracted all data and any disagreements were identified and resolved after reaching a consensus. The following data were extracted: (i) study characteristics (first author, year of publication, study design, study center, study period, number of patients for each group); (ii) patient characteristics (age in years, gender, Eastern Cooperative Oncology Group [ECOG] performance status, International Metastatic RCC Database Consortium [IMDC]/Heng risk score, Memorial Sloan Kettering Cancer Center [MSKCC]/Motzer risk score); (iii) tumor-related characteristics (histology, T stage and N stage according to TNM, number and location of metastases); (iv) treatment-related characteristics (type of targeted therapy); and (v) long-term survival outcomes (OS, PFS, and CSS).

### 2.4. Risk of Bias in Individual Studies

For observational studies, two independent reviewers (S.M.E. and I.A.Z) assessed the risk of bias using the Risk of Bias in Non-randomized Studies of Interventions (ROBINS-I) tool. The tool examines seven domains as a possible source of bias: (i) confounding, (ii) selection of participants (iii) classification of interventions, (iv) deviations from intended interventions, (v) missing data, (vi) measurement of outcomes, and (vii) selection of reported results. These domains are examined across three different levels (pre-intervention, at intervention, and post-intervention). For each domain, multiple standardized signaling questions are answered with “yes”, “probably yes”, “probably no”, “no”, and “no information”. Based on these answers, a domain-level judgement of bias is formulated and the risk of bias for each domain can be characterized as “low risk”, “moderate risk”, “high risk”, “critical risk”, or “no information”. Finally, an overall risk of bias judgement is made for each study using the same terms as for the domain-level judgments [20].

For RCTs, two independent reviewers (S.M.E. and I.A.Z) assessed the risk of bias using the Risk of Bias 2 (RoB 2) tool for randomized trials. The tool examines five domains as a possible source of bias: (i) randomization process, (ii) deviations from intended interventions, (iii) missing outcome data, (iv) measurement of the outcome, and (v) selection of the reported result. For each domain, multiple standardized signaling questions are answered with “yes”, “probably yes”, “probably no”, “no”, and “no information”. Based on these answers, a domain-level judgement of bias is formulated and the risk of bias for each domain can be characterized as “low risk”, “some concerns”, or “high risk”. Finally, an overall risk of bias judgement is made for each study using the same terms as for the domain-level judgments [21].

### 2.5. Statistical Analysis

#### 2.5.1. Data Pooling

Continuous variables were summarized as the means and standard deviations (SDs), and categorical variables were summarized as frequencies and percentages. We applied the methods described by Hozo et al. and Wan et al. to calculate means and SDs when continuous variables were reported as medians and ranges or medians and interquartile ranges, respectively [22,23]. All relative rates were calculated based on the available data for each variable of interest. All data were handled according to principles described in the Cochrane Handbook [24]. All time-to-event outcomes were summarized as hazard ratios (HRs) and 95% confidence intervals (CIs). Publication bias was assessed using funnel plots, which were examined for asymmetry. All statistical analyses were conducted with Stata IC 16.0 (StataCorp LLC, College Station, TX, USA) and R (Version 3.6.1) [25]. Statistical significance was set at 0.05 and all *p*-values were two tailed.

#### 2.5.2. Reconstruction of Individual Patient Survival Data

We used the methods described by Guyot et al. to reconstruct IPD from the survival curves of all eligible studies for the long-term survival outcomes (OS, PFS, CS) [26,27]. Vector and raster images of the Kaplan–Meier survival curves were pre-processed and digitized, so that the step function values and timing of steps could be extracted. Survival IPD were then reconstructed based on the numerical solutions to the inverted Kaplan–Meier product-limit equations. When the censoring pattern was not provided, we assumed that it was independent of failure time, and thus constant within each time interval [26]. Additional data, such as the number of patients at risk at every time interval or the total number of events, were used to further increase the accuracy of our calculations for the time-to-event data, when available [28]. Departures from monotonicity were detected using isotonic regression and corrected with a pool-adjacent-violators algorithm [26,27]. For every individual study, we compared summary statistics from our reconstructed IPD and curves (e.g., survival percentages at various time points, median survival time, total number of events, number-at-risk tables) with those reported in the original publications to ensure that they were accurate.

#### 2.5.3. One-Stage Survival Meta-Analysis

The Kaplan–Meier method was used to calculate the OS, PFS, and CSS. Both semi-parametric (i.e., Cox proportional hazards regression model) and non-parametric methods (i.e., restricted mean survival time [RMST]) were used to assess between-group difference.

The primary analysis for the OS, PFS, and CSS was performed using the Cox proportional hazards regression model, in which every patient within each individual study is assumed to be similarly failure prone to other patients belonging to that study. For these Cox models, the proportional hazards assumption was verified holistically from several assessments using the Grambsch–Therneau test for a non-zero slope, as well as by plotting scaled Schoenfeld residuals, log–log survival plots, and predicted versus observed survival functions. We plotted survival curves using the Kaplan–Meier product limit method and compared the HRs and 95% CIs of each group.

An alternative approach to analyzing time-to-event outcomes when non-proportional hazards are present is the RMST, which can be intuitively interpreted as the mean life expectancy up to a given time frame [29,30,31]. Accordingly, the life expectancy difference (LED) is the measure of the between-group RMST difference and expresses the absolute gain or loss in life expectancy, while the life expectancy ratio (LER) is the measure of the between-group RMST ratio and expresses the relative gain or loss in life expectancy [30]. We computed the RMST using the naïve Kaplan–Meier method which ignores study level effects, as it has been shown to always be unbiased in all meta-analytical scenarios [32].

#### 2.5.4. Two-Stage Survival Meta-Analyses

As a sensitivity analysis, we calculated summary HRs and 95% CIs for all individual studies based on the reconstructed IPD, and pooled them under the conventional “two-step” frequentist meta-analysis for all three long-term survival outcomes (OS, PFS, and CSS) [33]. We used the (DerSimonian–Laird) random-effects model to account for the significant clinical heterogeneity across the included studies, derived from factors such as the type of targeted therapy used, as well as the order of and the time intervals between CN and targeted therapy initiation. A subgroup analysis was performed depending on whether the included studies satisfied the proportional hazards assumption or not; the subgroup of studies that satisfied this criterion was considered to yield less biased pooled HR estimates. Between-study heterogeneity was assessed using the Cochran Q and the I^2^ statistic. High heterogeneity was defined with a significance level of *p* < 0.05 and a I^2^ value of ≥50%.

#### 2.5.5. Bayesian Meta-Analysis

We also performed a Bayesian one-stage random-effects meta-analysis to reflect our uncertainty regarding the potential survival benefits of CN, using an uninformative prior distribution β~Ν(0,10^10^), τ^2^~Γ(0.001,0.001). The analysis was performed in R using the spBayessurv package. We calculated the posterior median HRs and 95% credible intervals (CrIs) for each outcome (OS, PFS, and CSS) and compared them to the results of the one-stage frequentist meta-analysis.

As a second sensitivity analysis, we performed a two-stage Bayesian meta-analysis. We used the Tibshirani prior [34] as an uninformative prior and the half-normal prior (0,5) as a weakly informative prior. The analysis was performed in R using the rstanarm package. We then conducted a two-stage Bayesian meta-analysis using the random-effects model. We calculated the posterior median HRs and 95% credible intervals (CrIs) for each outcome (OS, PFS, and CSS) and compared them to the results of the two-stage frequentist meta-analysis.

#### 2.5.6. Subgroup Analysis According to Cytoreductive Nephrectomy Timing

A subgroup analysis was initially planned to compare outcomes between upfront and deferred CN, due to the ongoing debate regarding the optimal timing of CN. However, none of the included studies specifically reported outcomes on deferred CN. Therefore, we limited our subgroup analysis to studies reporting on upfront CN, after excluding studies with mixed (upfront and deferred) CN groups. We performed both one-stage frequentist and Bayesian meta-analyses using the same methodology as for the primary analyses described above.

## 3. Results

### 3.1. Study Selection and Characteristics

After removing all duplicates, we identified 998 unique articles through our systematic search. We determined that 96 articles were relevant based on their titles and abstracts, and further evaluated their full texts for eligibility. Fifteen studies (fourteen retrospective cohort studies and one RCT) fulfilled the pre-determined inclusion criteria and were included in our meta-analysis (Figure 1) [15,16,17,35,36,37,38,39,40,41,42,43,44,45,46]. On one occasion, data from two studies with overlapping populations reporting on different outcomes were combined [39,47]. A total of 2234 patients received CN combined with targeted treatment, while 1756 patients received targeted therapy alone. Detailed study characteristics and patient demographics are shown in Table 1, clinical characteristics and type of targeted therapy used in Table 2.

### 3.2. Risk of Bias Assessment Individual Patient Data and Survival Curve Reconstruction

We assessed the individual risk of bias of 14 observational studies using the ROBINS-I tool. The overall risk of bias was determined to be low in one study [41], moderate in twelve studies [15,17,35,36,37,38,39,40,42,43,44,46] and serious in one study [45]. No studies were found to be at critical risk of bias (Figure 2A,B).

We assessed the individual risk of bias of one RCT [16] using the RoB2 tool. The overall risk of bias was determined to be high, stemming from deviations from the intended interventions, while some concerns were present regarding the randomization process (Figure 2C,D).

### 3.3. Individual Patient Data and Survival Curve Reconstruction

The Kaplan–Meier curves for each outcome (OS, PFS, and CSS) were appropriately processed and digitized. A total of sixteen OS curves, seven PFS curves, and three CSS curves were reconstructed. A side-by-side comparison of our reconstructed Kaplan–Meier curves and those found in the original publications is provided in Appendix A. Using a previously validated methodology, we recuperated IPD from the survival curves of each outcome (Appendix A).

### 3.4. One-Stage Frequentist Survival Meta-Analysis

We used the Cox proportional hazards model for our main analysis of all outcomes (OS, PFS, and CSS), since we did not detect any violation of the proportionality-of-hazards assumption upon a holistic assessment using the Grambsch–Therneau test and by visualizing scaled Schoenfeld residuals, log–log survival plots, and predicted versus observed survival curves (Appendix A). We nevertheless carried out secondary analyses with non-parametric methods (i.e., RMST), according to our protocol, even though the proportionality-of-hazards assumption was not rejected.

#### 3.4.1. Overall Survival

The OS curve of the pooled patient cohorts either receiving CN plus targeted therapy (*n* = 2205) or targeted therapy alone (*n* = 1752) derived from fifteen studies [15,16,17,35,36,37,38,40,41,42,43,44,45,46] is presented in Figure 3A. The median OS was 23.3 months (95% CI: 22.0 to 24.9) in the CN group and 12.9 months (95% CI: 12.0 to 14.1) in the non-CN group. Patients receiving the combination of CN and targeted therapy had significantly lower risk of death compared to those receiving targeted therapy alone (HR: 0.58, 95% CI: 0.54 to 0.62, *p* < 0.0001).

In the RMST analysis, the LED was 6.0 months (95% CI: 5.2 to 6.8, *p* < 0.0001) at 3 years and 9.4 months (95% CI: 8.1 to 10.7, *p* < 0.0001) at 5 years, both favoring the CN plus targeted therapy patient cohort. Accordingly, the LER was 1.36 (95% CI: 1.30 to 1.42, *p* < 0.0001) at 3 years and 1.48 (95% CI: 1.40 to 1.56, *p* < 0.0001) at 5 years, again favoring the cohort receiving CN plus targeted therapy (Table 3).

#### 3.4.2. Progression-Free Survival

The PFS curve of the pooled patient cohorts either receiving CN plus targeted therapy (*n* = 608) or targeted therapy alone (*n* = 642) derived from seven studies [16,17,37,43,45,46,47] is presented in Figure 3B. The median PFS was 8.4 months (95% CI: 7.3 to 9.6) in the CN group and 8.5 months (95% CI: 6.5 to 9.0) in the non-CN group. The results were not compatible with clinically meaningful differences between the two groups (HR: 0.90, 95% CI: 0.80 to 1.02, *p* = 0.09).

In the RMST analysis, the LED was 1.1 months (95% CI: [−0.2] to [2.3], *p* = 0.10] at 3 years and 1.4 months (95% CI: [−0.5] to [3.3], *p* = 0.15) at 5 years. Accordingly, the LER was 1.09 (95% CI: 0.98 to 1.20, *p* = 0.10) at 3 years and 1.10 (95% CI: 0.97 to 1.25, *p* = 0.15) at 5 years, which was not compatible with clinically meaningful differences (Table 3).

#### 3.4.3. Cancer-Specific Survival

The CSS curve of the pooled patient cohorts either receiving CN plus targeted therapy (*n* = 313) or targeted therapy alone (*n* = 278) derived from three studies [15,17,35] is presented in Figure 3C. The median CSS was 32.3 months (95% CI: 27.0 to 37.0) in the CN group and 17.6 months (95% CI: 12.7 to 20.9) in the non-CN group. Patients receiving the combination of CN and targeted therapy had significantly lower risk of death from mRCC compared to those receiving targeted therapy alone (HR: 0.63, 95% CI: 0.53 to 0.75, *p* < 0.0001).

In the RMST analysis, the LED was 6.2 months (95% CI: 4.2 to 8.3, *p* < 0.0001) at 3 years and 9.4 months (95% CI: 6.1 to 12.8, *p* < 0.0001) at 5 years, both favoring the cohort receiving CN plus targeted therapy. Accordingly, the LER was 1.32 (95% CI: 1.20 to 1.46, *p* < 0.0001) at 3 years and 1.39 (95% CI: 1.23 to 1.57, *p* < 0.0001) at 5 years, again favoring the cohort receiving CN plus targeted therapy (Table 3).

### 3.5. Two-Stage Frequentist Survival Meta-Analysis

In the two-stage frequentist meta-analysis, CN combined with targeted therapy was associated with superior OS (HR: 0.59, 95% CI: 0.49 to 0.71, *p* < 0.001, I^2^ = 79.31%) compared to targeted therapy alone. In the subgroup analysis, the combination of CN and targeted therapy was associated with superior OS in the subgroup satisfying the proportionality-of-hazards assumption (HR: 0.50, 95% CI: 0.44 to 0.58, *p* < 0.001, I^2^ = 29.60%), but was inconclusive in the subgroup violating the proportionality-of-hazards assumption (HR: 0.82, 95% CI: 0.59 to 1.13, *p* = 0.23, I^2^ = 81.10%). The PFS results were inconclusive when all studies (HR: 0.87, 95% CI: 0.69 to 1.09, *p* = 0.22, I^2^ = 65.77%) or only those satisfying the proportionality-of-hazards assumption (HR: 0.82, 95% CI: 0.62 to 1.09, *p* = 0.18, I^2^ = 70.38%) were included. The results were inconclusive regarding CSS between the two groups (HR: 0.49, 95% CI: 0.20 to 1.24, *p* = 0.13, I^2^ = 94.70%) (Appendix A).

### 3.6. Bayesian Meta-Analysis

In the one-stage Bayesian meta-analysis, CN combined with targeted therapy was associated with superior OS (HR: 0.59, 95% CrI: 0.55 to 0.63) and CSS (HR: 0.63, 95% CrI: 0.53 to 0.75). No meaningful clinical difference was detected in PFS between the two groups (HR: 0.91, 95% CrI: 0.80 to 1.02) (Table 3).

In the two-stage Bayesian meta-analysis, CN combined with targeted therapy was associated with superior OS (both priors: HR: 0.59, 95% CrI: 0.48 to 0.72). No statistically significant difference was detected in PFS (both priors: HR: 0.87, 95% CrI: 0.64 to 1.14) and CSS (informative prior: HR: 0.51, 95% CrI: 0.21 to 1.16 / uninformative prior: HR: 0.50, 95% CrI: 0.08 to 2.91) of the two groups (Appendix A).

### 3.7. Subgroup Analysis According to Cytoreductive Nephrectomy Timing

Eleven studies were included for the OS [16,17,35,36,37,38,39,40,44,45,46], six studies for the PFS [16,17,37,45,46,47], and two studies for the CSS subgroup analysis [17,35] comparing upfront CN followed by targeted therapy vs. targeted therapy alone. Patients undergoing upfront CN had superior OS compared to those who did not, as shown in both one-stage frequentist (HR: 0.70, 95% CI: 0.63 to 0.78, *p* < 0.001) (Figure 4A) and Bayesian (HR: 0.70, 95% CrI: 0.64 to 0.78) meta-analyses. No clinically meaningful differences were found between the two groups in the PFS subgroup analysis with either one-stage frequentist (HR: 0.94, 95% CI: 0.83 to 1.07, *p* = 0.33) (Figure 4B) or Bayesian (HR: 0.94, 95% CrI: 0.84 to 1.07) meta-analysis. The CSS subgroup analysis was inconclusive in both one-stage frequentist (HR: 0.97, 95% CI: 0.75 to 1.25, *p* = 0.81) (Figure 4C) and Bayesian (HR: 0.97, 95% CrI: 0.75 to 1.23) meta-analysis. However, the proportional hazards assumption was violated, as shown by the Grambsch–Therneau test (*p* = 0.0058). Therefore, the non-parametric (RMST) analysis was considered more reliable in this case. Upfront CN was associated with superior CSS at 3 years (LED: 3.4 months, 95% CI: 4.2 to 8.3, *p* < 0.0001, LER: 1.16, 95% CI: 1.02 to 1.33) but not at 5 years (LED: 3.4 months, 95% CI: [−1.7] to [8.5], LER: 1.12, 95% CI: 0.95 to 1.33) (Appendix A).

## 4. Discussion

In this systematic review and IPD meta-analysis, we showed that the combination of CN and targeted therapy in mRCC is associated with superior long-term survival outcomes compared to targeted therapy alone. Both OS and CSS were superior in the group receiving CN. In contrast, no clinically meaningful differences were detected in the PFS between the groups.

Although several meta-analyses have attempted to address this topic in the past, limitations in their methodology have precluded any definitive conclusions [48,49,50,51,52]. In some cases, multiple overlapping populations were included [48,49,50,51,52] and eligible studies were omitted [48,50,51,52] resulting in significant bias due to the disproportionate representation of certain patient populations. An additional limitation of studies analyzing databases (e.g., the Surveillance Epidemiology and End-Results database and the National Cancer Data Base) included in the previous meta-analyses [48,50,52] is the use of surrogate coding markers, such as the year of diagnosis or the receipt of systemic therapy, to identify patients receiving targeted therapy [53,54]. Therefore, these studies may have included a significant proportion of patients that did not actually receive targeted therapy. To avoid these shortcomings, we ensured that every population was represented only once in our meta-analysis and we only included studies with confirmed targeted therapy use in their intention-to-treat population. To our knowledge, this is the first meta-analysis on the topic to incorporate IPD, which is considered the “gold-standard” method to meta-analyze time-to-event outcomes, and the first to investigate additional survival outcomes other than OS, such as CSS [55]. The higher precision achieved by using IPD compared to aggregate study level data is showcased by the inconclusive or less precise results of our two-stage meta-analyses, which emulate aggregate data meta-analyses, compared with the much more precise results of our primary IPD meta-analyses with both frequentist and Bayesian approaches.

Even though CN has been a staple in the treatment of mRCC for decades, the exact mechanism behind its survival benefit remains unclear. A variety of theories have been proposed over the years. An intuitive explanation is that CN reduces the overall tumor load and thus prolongs the period needed for it to reach lethal levels [14]. There is also evidence suggesting that CN further opposes tumor growth by indirectly affecting the tumor microenvironment. Removing functional nephrons induces a mild systemic metabolic acidosis, which may be enough to overwhelm the acid-base regulation ability of tumor cells, resulting in necrosis [56]. Similarly, many angiogenic factors that promote tumor growth, such as VEGF, decrease following nephrectomy [57]. The interaction between CN and the immune system remains a point of debate. The ability of mRCC to downregulate the immune system through various pathways is well established, and therefore removing the primary tumor may enhance the immune response against the remaining cancer cells [58,59]. However, this effect may be counterbalanced by the ongoing systemic inflammation, which promotes tumorigenesis, as well as the immunosuppressive effects of the surgery itself [59,60]. Although the concurrent presence of widespread inflammation and immunosuppression may initially appear as counterintuitive, this interaction is particularly illustrated through C-reactive protein, a marker of inflammation. Indeed, C-reactive protein levels closely correlate with the tumor-induced immunosuppression in mRCC [61]. This finding suggests that the tumor-induced immunosuppression and widespread inflammation are intertwined as part of the generalized immune dysregulation caused by mRCC. CN may be able to partially reverse these effects in some but not all patients. For instance, there is a growing body of evidence suggesting that CN may be less beneficial to patients with generalized inflammation reflected by high C-reactive protein levels [37].

An important consideration when examining the benefits of CN is appropriate patient selection [62]. Being an invasive procedure on an already disease-burdened patient population, CN is associated with significant morbidity and mortality that is higher compared with that of standard nephrectomy [63]. Those with poor baseline characteristics may thus never fully recover from the operation to receive targeted therapy or inevitably experience rapid tumor progression in the immediate post-operative period [64]. This phenomenon may be explained by the direct and indirect effects of a major abdominal operation such as CN, namely the surgery-induced state of immunosuppression and release of growth factors, as well as the potential delay of systemic therapy initiation related to surgical complications, respectively [65]. As a result, the survival benefit in patients with poor prognostic factors is minimal, while the operation itself and the high rate of post-operative complications may significantly impact quality of life [15,42]. Current guidelines regarding CN reflect these concerns and heavily question CN’s current role in mRCC management [66]. Consequently, studies showing a survival benefit with CN may be biased in their patient selection. In our pooled sample, the non-CN group had significantly higher proportion of patients with poor IMDC risk score, Karnofsky score <80%, T3/T4 stage, N+ stage, and >2 metastatic sites, all of which may have confounded the results. In contrast, the ECOG scores among the two groups were similar, while the CN group had more patients with poor MSKCC score. This contradictory finding highlights the lack of a uniform score scale to evaluate the patients’ baseline surgical risk. Even though numerous prognostic scales have been developed for this purpose during the cytokine era, external validation has shown that they all perform poorly in a targeted therapy-predominant cohort [62]. For this reason, newer scales have been developed during the targeted therapy era; examples include those by McIntosh et al. [67] and Marchioni et al. [68]. However, these remain to be prospectively evaluated and externally validated before they can be implemented into routine clinical practice. As shown in our study, authors resort to a variety of prognostic scales as a substitute to stratify their patients, reflecting the lack of a specialized prognostic model to satisfy this need. This approach may be problematic as these scales were designed for different purposes and are not directly comparable with each other [69]. A post-hoc analysis of the CARMENA trial particularly highlights this concern by suggesting that patients with one but not two IMDC risk factors (both classified as intermediate-risk patients) benefited from CN [70]. Other authors have used their own models to stratify patients based on prognostic factors derived from regression analyses [36,44]. Regardless of the approach, several studies have shown that CN offered a considerable survival advantage, even when accounting for these risk factors by performing subgroup analyses in patients with more favorable prognosis [15,36,42,44]. Even though we were not able to synthesize their results due to the heterogeneity in the stratification method used, this evidence suggests that the benefit of CN stands even after taking selection bias into consideration.

Apart from appropriate patient selection, the timing of CN relative to targeted therapy initiation is another parameter that may affect treatment outcomes. The SURTIME trial comparing upfront and deferred CN hinted that the latter may lead to an OS advantage but failed to provide a definitive answer, presumably due to poor accrual [71]. A study pooling data from multiple trials found an OS advantage of deferred over the upfront approach [72]. A more recent multi-institutional study using real-world data also came to the same conclusion [73]. In contrast, a study using NCDB data suggested an advantage of upfront CN [54]. In our meta-analysis, we were unable to directly compare outcomes between the two approaches, as none of the included studies utilized deferred CN exclusively. After excluding studies with both upfront and deferred CN in their sample, we found that patients undergoing upfront CN followed by targeted therapy still had superior OS and CSS compared to those receiving targeted therapy alone. However, the benefit was smaller compared to the primary analysis that also included deferred CN cases. This finding is in accordance with previous studies suggesting that deferred CN may be optimal in terms of timing, but also shows that upfront CN in select patients may still lead to better outcomes compared to no CN.

The main strength of this study is its robust methodology and large sample size, particularly for the OS analysis. Nonetheless, our results should be interpreted with caution due to the inherent limitations of our study. First, most of the included studies were retrospective in nature. In the context of our study, this is important because it may facilitate immortal time bias [74]. Most included studies used either the time of mRCC diagnosis [15,38,39] or the time of targeted therapy initiation [42,43,45,46] as the starting point in their survival analysis. Therefore, patients receiving deferred CN may have been considered “immortal” up to the point of undergoing CN, while those who were scheduled to receive CN but died before doing so may have been excluded from the CN group, thus skewing the results in favor of CN. In contrast, when patients received upfront CN and the date of targeted therapy initiation was used as the starting point, the results may have been skewed towards the opposite direction as the non-CN group is considered “immortal” during the period from CN and its postoperative recovery until targeted therapy initiation [40]. Second, due to the inability to obtain IPD for variables other than survival outcomes, we were not able to perform subgroup analyses for factors with prognostic significance that may influence patient selection for CN. Examples include clear-cell vs. non-clear-cell mRCC [17] and favorable vs. intermediate vs. poor IMDC or MSKCC risk score [42]. Therefore, the results of the crude cohorts may impart a degree of inherent selection bias. Third, some studies included a small proportion of patients that did not receive the planned targeted therapy [44], or subsequently received other forms of systemic therapy, such as immunotherapy [16]. We decided to include these studies regardless, as we deemed the increase in study power to be more important than the slight increase in heterogeneity by deviation from the intended protocol. Lastly, as with any systematic review, some of the articles did not report on all variables of interest, and thus all relative rates were calculated according to the availability of data.

## 5. Conclusions

The combination of CN and targeted therapy for mRCC may lead to superior long-term survival outcomes compared to targeted therapy alone. Careful patient selection based on baseline prognostic factors is required to achieve optimal survival for patients with mRCC.

## Figures and Tables

**Figure 1 cancers-13-00695-f001:**
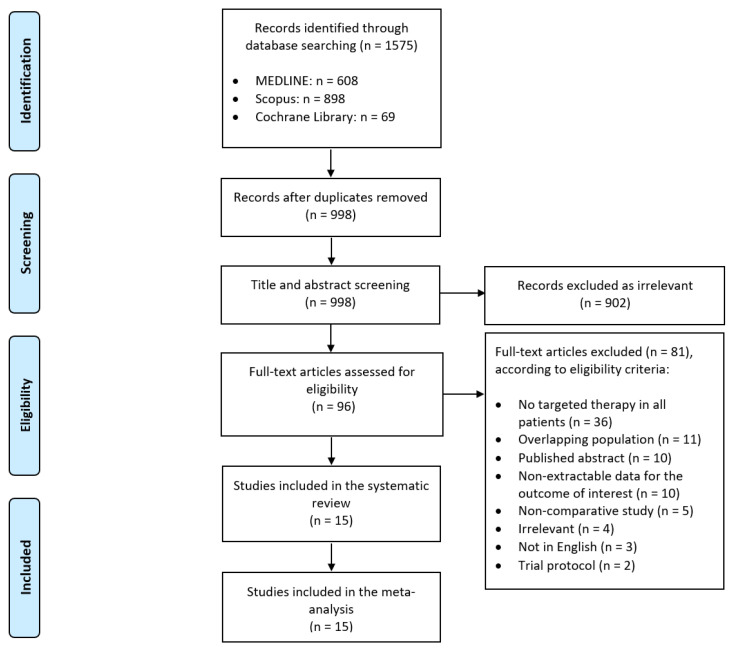
Preferred Reporting Items for Systematic Reviews and Meta-Analyses (PRISMA) flow diagram of the study selection process.

**Figure 2 cancers-13-00695-f002:**
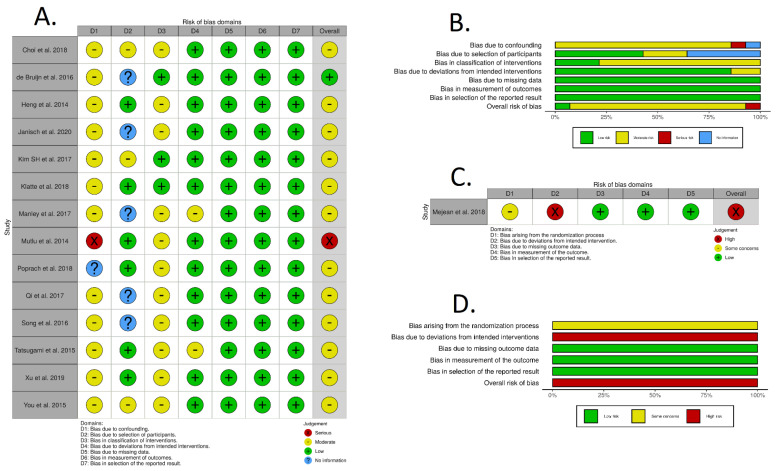
Risk of bias assessment for observational studies using the ROBINS-I tool (**A**,**B**) and randomized controlled trials using the RoB2 tool (**C**,**D**).

**Figure 3 cancers-13-00695-f003:**
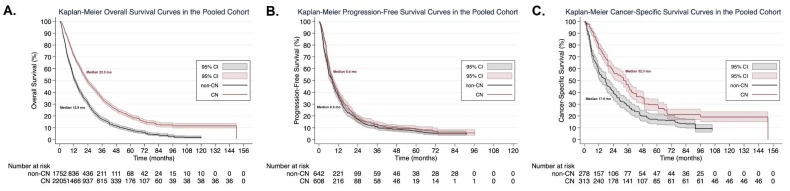
Kaplan–Meier plots of the pooled cohorts for overall survival (**A**), progression-free survival (**B**), and cancer-specific survival (**C**), derived from the one-stage frequentist random-effects survival meta-analysis using reconstructed individual patient data.

**Figure 4 cancers-13-00695-f004:**
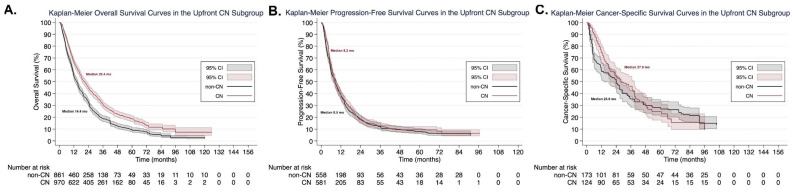
Kaplan–Meier plots in the upfront cytoreductive nephrectomy (CN) subgroup for overall survival (**A**), progression-free survival (**B**), and cancer-specific survival (**C**), derived from the one-stage frequentist random-effects survival meta-analysis using reconstructed individual patient data.

**Table 1 cancers-13-00695-t001:** Study characteristics and patient demographics.

Author	Year	Center and Country	Study Period	CN Patients	Non-CN Patients	Age (years)		Male Sex		Clear Cell Histology	
						CN	Non-CN	CN	Non-CN	CN	Non-CN
Choi et al. [15]	2018	Samsung Medical Center, Seoul, South Korea	January 2005 to December 2015	189	105	60.7 ± 14.2	56.3 ± 9.8	149	82	166	76
de Bruijn et al. [41]	2016	The Netherlands Cancer Institute, Amsterdam, The Netherlands	January 2006 to December 2012	39	29	NA	NA	NA	NA	39	29
Heng et al. [42]	2014	IMDC database (20 international centers)	NA	982	676	59.8 ± 10.9	61.5 ± 11.4	721	488	841	450
Janisch et al. [17]	2020	University Medical Center Hamburg-Eppendorf, Hamburg, Germany	2000 to 2016	104	158	60.7 ± 9.8	61.3 ± 11.2	80	113	89	130
Kim et al. [43]	2017	Research Institute and Hospital of National Cancer Center, Goyang, South Korea	January 2000 to December 2015	27	84	NA	NA	NA	NA	NA	NA
Klatte et al. [40]	2018	Cambridge Oncology Registry, UK	2006 to 2017	97	164	58.6 ± 12.4	64.4 ± 9.9	65	116	80	123
Manley et al. [44]	2017	Washington University School of Medicine, Division of Urology, St. Louis, Missouri, USA	2005 to 2013	88	35	57.4 ± 10.4	57.8 ± 10.4	NA	NA	NA	NA
Mejean et al. [16]	2018	Multicenter (79 centers from France, Norway, England, Scotland, Sweden)	September 2009 to September 2017	226	224	60.7 ± 8.5	60.2 ± 9.5	169	167	226	224
Mutlu et al. [45]	2014	Akdeniz University, Antalya, Afyon Kocatepe University, Afyon and Medipol University, Istanbul, Turkey	NA	28	24	53.6 ± 9.8	67.5 ± 10.5	22	16	NA	NA
Poprach et al. [46]	2018	Renal Cell Carcinoma Information System (RENIS) registry, Czech Republic	August 2011 to December 2015	114	71	NA	NA	NA	NA	NA	NA
Qi et al. [35]	2017	Peking University First Hospital, Institute of Urology, Beijing, China	April 2008 to October 2014	20	15	NA	NA	15	10	16	12
Song et al. [36]	2016	Cancer Hospital (Institute), Chinese Academy of Medical Sciences, Beijing, China	NA	51	23	NA	NA	37	19	NA	NA
Tatsugami et al. [38]	2015	7 centers from Japan	January 2001 to December 2010	103	25	NA	NA	NA	NA	NA	NA
Xu et al [37].	2019	Fudan University Shanghai Cancer Center (FUSCC), Shanghai, China	May 2009 to June 2018	70	48	NA	NA	46	38	55	37
You et al. [39]	2015	Asan Medical Center, University of Ulsan College of Medicine, Seoul, South Korea	2006 to 2012	96	75	56.5 ± 10.4	60.2 ± 12.8	66	51	92	64
Total			2000–2018	2234	1756	59.6 ± 11.1	61.2 ± 11.0	1370/1853 (73.9)	1100/1512 (72.8)	1604/1793 (89.5)	1145/1334 (85.8)

Values are expressed as the means ± standard deviations or as frequencies (percentages). CN: cytoreductive nephrectomy; NA: not available.

**Table 2 cancers-13-00695-t002:** Clinical characteristics of all patients included in the analysis.

Clinical Characteristic	CN Group	Non-CN Group
IMDC Risk Score		
0	81/1108 (7.3)	8/910 (0.9)
1–2	720/1108 (65.0)	459/910 (50.4)
3–6	307/1108 (27.7)	443/910 (48.7)
MSKCC Risk Score		
0	19/384 (5.0)	48/490 (9.8)
1–2	232/384 (60.4)	308/490 (62.9)
3–6	133/384 (34.6)	134/490 (27.4)
ECOG ≥ 2	22/539 (4.1)	19/502 (3.8)
Karnofsky ≥ 80%	976/1187 (82.2)	606/980 (61.8)
T1/2 Stage	365/901 (40.5)	174/490 (35.5)
N1 Stage	149/372 (40.1)	121/233 (51.9)
>2 Metastases	76/321 (23.7)	136/393 (34.6)
Brain Metastases	98/1221 (8.0)	118/1137 (10.4)
Bone Metastases	565/1519 (37.2)	622/1388 (44.8)
Liver Metastases	239/1326 (18.0)	274/1178 (23.3)
Lung Metastases	426/566 (75.3)	504/698 (72.2)
Lymph Node Metastases	188/418 (45.0)	276/543 (50.8)
Type of Targeted Therapy		
Sunitinib	1110/1572 (70.6)	1029/1249 (82.4)
Pazopanib	166/1351 (12.3)	108/1019 (10.6)
Axitinib	31/1219 (2.5)	32/921 (3.5)
Sorafenib	285/1463 (19.5)	110/1172 (9.4)
Famitinib	5/51 (9.8)	9/23 (39.1)
Bevacizumab	42/965 (4.4)	10/673 (1.5)
Everolimus	34/1191 (2.9)	45/897 (5.0)

Values are expressed as the means ± standard deviations or as frequencies (percentages). ECOG: Eastern Cooperative Oncology Group; IMDC: International Metastatic RCC Database Consortium; MSKCC: Memorial Sloan Kettering Cancer Center; mets: metastases; CN: cytoreductive nephrectomy; NA: not available.

**Table 3 cancers-13-00695-t003:** Primary analysis of overall, progression-free, and cancer-specific survival estimates using reconstructed survival information.

One-Stage Meta-Analysis		Overall Survival		Progression-Free Survival		Cancer-Specific Survival	
		Relative Effect of CN versus Non-CN (95% Cl/CrI)	*p*-Value for Relative Effect	Relative Effect of CN versus Non-CN (95% Cl/CrI)	*p*-Value for Relative Effect	Relative Effect of CN versus Non-CN (95% Cl/CrI)	*p*-Value for Relative Effect
Frequentist Approach	Cox Proportional Hazards Model	0.58 ^a^ (0.54–0.62)	<0.0001	0.90 ^b^ (0.80–1.02)	0.093	0.63 ^c^ (0.53–0.75)	<0.0001
	Life Expectancy Difference (up to 3 years)	6.0 months (5.2–6.8)	<0.0001	1.1 months [(−0.2)–(2.3)]	0.100	6.2 months (4.2–8.3)	<0.0001
	Life Expectancy Ratio (up to 3 years)	1.36 (1.30–1.42)	<0.0001	1.09 (0.98–1.20)	0.100	1.32 (1.20–1.46)	<0.0001
	Life Expectancy Difference (up to 5 years)	9.4 months (8.1–10.7)	<0.0001	1.4 months [(−0.5)–(3.3)]	0.150	9.4 months (6.1–12.8)	<0.0001
	Life Expectancy Ratio (up to 5 years)	1.48 (1.40–1.56)	<0.0001	1.10 (0.97–1.25)	0.150	1.39 (1.23–1.57)	<0.0001
Bayesian Approach	Cox Proportional Hazards Model	0.59 (0.55–0.63)	N/A	0.91 (0.80–1.02)	N/A	0.63 (0.53–0.75)	N/A

CN: cytoreductive nephrectomy; CI: confidence interval; CrI: credibility interval; N/A: not available. ^a^ Assessing for non-proportional hazards *p*-value = 0.074, from the Grambsch–Therneau test. ^b^ Assessing for non-proportional hazards *p*-value = 0.260, from the Grambsch–Therneau test. ^c^ Assessing for non-proportional hazards *p*-value = 0.150, from the Grambsch–Therneau test.

## Data Availability

The recuperated individual patient data for the overall survival, progression-free survival, and cancer-specific survival outcomes are provided on Appendix A. All additional data presented in this study are available on request from the corresponding author.

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
