# Peer review of "Long-Term Survival Outcomes of Cytoreductive Nephrectomy Combined with Targeted Therapy for Metastatic Renal Cell Carcinoma: A Systematic Review and Individual Patient Data Meta-Analysis"

_cancers, 2021, doi:10.3390/cancers13040695_

Round 1

Reviewer 1 Report

1) General comments

The authors performed individual patient data (IPD) meta-analyses to asses cytoreductive nephrectomy (CN) for patients with metastatic renal cell carcinoma. One-stage meta-analysis showed benefit of overall survival (OS) and cancer-specific survival (CSS). The OS benefit was also noted in the two-stage meta-analysis. They also performed subgroup analysis according to CN timing. They excluded studies with mixed (upfront and deferred) CN group and showed the benefit of OS in upfront CN group. This is well written and the information would be helpful. The following points should be revised and clarified to improve the manuscript.

2) Specific comments for revision

a) major

  1. CN should be performed in mRCC patients with good and intermediate prognosis, because high rate of postoperative mortality has been reported. I recommend the authors to analyze the immediate postoperative mortality.
  2. In Figure 3 and 4, the authors presented Kaplan-Meier curves of OS, PFS, and CSS. I recommend the authors to show median time of each curves.
  3. Rapid disease progression reported as hyperprosgression is very important to consider CN atter targeted therapy. I recommend the authors to discuss.

b) minor

  1. The authors should revise the word “metastatic mRCC” to “mRCC” in line 394.

Author Response

Reviewer: "The authors performed individual patient data (IPD) meta-analyses to asses cytoreductive nephrectomy (CN) for patients with metastatic renal cell carcinoma. One-stage meta-analysis showed benefit of overall survival (OS) and cancer-specific survival (CSS). The OS benefit was also noted in the two-stage meta-analysis. They also performed subgroup analysis according to CN timing. They excluded studies with mixed (upfront and deferred) CN group and showed the benefit of OS in upfront CN group. This is well written and the information would be helpful. The following points should be revised and clarified to improve the manuscript.”

Response: Thank you very much for your feedback. We hope that after following your suggestions, our revisions address your concerns and add to the clarity of our manuscript.

Reviewer: "CN should be performed in mRCC patients with good and intermediate prognosis, because high rate of postoperative mortality has been reported. I recommend the authors to analyze the immediate postoperative mortality.”

Response: Thank you for your comment. As our meta-analysis focused on the long-term outcomes of cytoreductive nephrectomy, none of the included studies reported any data on CN’s postoperative mortality. We have made sure to discuss this topic in the Discussion section of our revised manuscript (lines 447-450).

Reviewer: "In Figure 3 and 4, the authors presented Kaplan-Meier curves of OS, PFS, and CSS. I recommend the authors to show median time of each curves."

Response: We have now accordingly added the median survival time for each group in the Results section (lines 312-313, 330-331, and 342-343). We have also updated our figures (Figures 3 and 4) to clearly display the median values next to the curve of each group. To further improve the information provided by our figures, we have now also added the number of patients at risk for each time interval, as well as the 95% confidence intervals for each curve.

Reviewer: "Rapid disease progression reported as hyperprogression is very important to consider CN atter targeted therapy. I recommend the authors to discuss."

Response: We have revised our Discussion section to address this topic in lines 452-456.

Reviewer: "The authors should revise the word “metastatic mRCC” to “mRCC” in line 394.”

Response: Thank you for catching this typographical error. We have amended it in the revised version.

Reviewer 2 Report

I have reviewed the manuscript "Long-term survival outcomes of cytoreductive nephrectomy combined with targeted therapy for metastatic renal cell carcinoma: a systematic review and individual patient data meta-analysis". The study subject is relevant, addressing an inconclusive issue in the field. The manuscript is very well-written, the methodology is correct is clearly described. The conclusions are supported by the results. 

My only comment is whether the authors explored potential interaction by relevant effect-modifiers. If done, please state so; if not, please add such analyses.

Author Response

Reviewer: "I have reviewed the manuscript "Long-term survival outcomes of cytoreductive nephrectomy combined with targeted therapy for metastatic renal cell carcinoma: a systematic review and individual patient data meta-analysis". The study subject is relevant, addressing an inconclusive issue in the field. The manuscript is very well-written, the methodology is correct is clearly described. The conclusions are supported by the results. My only comment is whether the authors explored potential interaction by relevant effect-modifiers. If done, please state so; if not, please add such analyses.

Response: Thank you very much for your feedback. Although we originally planned to perform additional analyses to explore the effect of such interactions, we were unable to do so as the authors of the original studies used different score scales to report on preoperative variables that could act as confounders or effect modifiers to our results. We now extensively discuss this issue in the revised Discussion section of our manuscript (lines 447-485) and comment on the findings of individual studies included in our analyses that adjusted their results for these factors.

Reviewer 3 Report

The authors performed a systematic review to gather patient data to compare the long-term survival outcomes of cytoreductive nephrectomy (CN) combined with targeted therapy vs targeted therapy alone in patients with metastatic renal cell carcinoma. The authors defined targeted therapy as systemic therapy with vascular endothelial growth (VEGF) receptor-directed tyrosine kinase inhibitors (TKIs), anti-VEGF monoclonal antibodies, or mammalian target of rapamycin inhibitors. Both OS and CSS were superior in the group receiving CN. No clinically meaningful differences were detected in the PFS between the groups.

The authors concluded that the combination of CN and targeted for metastatic renal cell carcinoma may lead to superior long-term survival outcomes compared to targeted therapy alone.

This is an excellent review, with a robust methodology and a large sample size.

The discussion is very well presented and discuss the limitations of this study.

The study is very well detailed.

Author Response

Reviewer: "The authors performed a systematic review to gather patient data to compare the long-term survival outcomes of cytoreductive nephrectomy (CN) combined with targeted therapy vs targeted therapy alone in patients with metastatic renal cell carcinoma. The authors defined targeted therapy as systemic therapy with vascular endothelial growth (VEGF) receptor-directed tyrosine kinase inhibitors (TKIs), anti-VEGF monoclonal antibodies, or mammalian target of rapamycin inhibitors. Both OS and CSS were superior in the group receiving CN. No clinically meaningful differences were detected in the PFS between the groups. The authors concluded that the combination of CN and targeted for metastatic renal cell carcinoma may lead to superior long-term survival outcomes compared to targeted therapy alone. This is an excellent review, with a robust methodology and a large sample size. The discussion is very well presented and discuss the limitations of this study. The study is very well detailed.”

Response: Thank you very much for your kind feedback.